# Study on the Influences of Inter-Governmental Information Flow and Interdepartmental Collaborative Supervision on Pork Quality: A Case Study in China

**DOI:** 10.3390/foods13152387

**Published:** 2024-07-28

**Authors:** Linhai Wu, Zhiyu Chen

**Affiliations:** School of Business, Institute for Food Safety Risk Management, Jiangnan University, Wuxi 214122, China; chenzhiyu0912@163.com

**Keywords:** pork supply chain, social co-governance, interdepartmental collaborative governance, coordination degree, evolutionary game

## Abstract

To study the influences of the degree of interdepartmental collaborative supervision on the behavioral strategy selection of pig farmers, pig slaughterers, and pork processing manufacturers in the pork supply chain system, this study established a three-party evolutionary game model involving pig farmers, pig slaughterers, and pork processing manufacturers based on the social co-governance framework by focusing on the interdepartmental information sharing mechanism and cooperative governance. Here, we examined how the degree of collaborative supervision among government departments influences the behavioral strategy selection of these parties by focusing on key mechanisms such as information sharing and interdepartmental collaborative governance. Our findings revealed that within a social co-governance system, the strategic choices of the three entities in the pork supply chain closely correlate with the coordination level of collaborative supervision among government departments, particularly through information-sharing mechanisms. Additionally, the strategies are influenced by market-based contract supervision among entities, consumer reporting intensity, and the collaborative governance capabilities of the government, market actors, and consumers. Higher levels of social co-governance are associated with fewer risky links in the pork supply chain and reduced overall risk. Key factors affecting the behavioral strategy selection of the subjects in the pork supply chain include interdepartmental collaborative governance among government departments (e.g., optimizing random inspection frequencies, adjusting economic penalties, and disclosing enterprise market credit information via information sharing mechanisms), consumer complaint probabilities, and the intensity of mutual supervision among enterprises. Therefore, to enhance pork supply chain quality and mitigate risks, it is crucial to enhance the coordination of collaborative supervision among government departments, encourage consumer reporting, and improve market-based mutual supervision mechanisms among upstream and downstream subjects in the supply chain.

## 1. Introduction

Despite the recent scientific and technological breakthroughs in the food industry to enhance food safety, global food safety issues remain a frequent and ongoing occurrence. Each year, approximately 600 million people become ill from consuming contaminated food, leading to 420,000 deaths and the loss of 33 million healthy life years (years lived with disability, YLD) [1,2]. Therefore, food safety remains a significant public concern worldwide [3]. Since the onset of industrialization, many countries have made continuous efforts to manage food safety risks. Developed countries, such as the United States, have successfully implemented social co-governance and interdepartmental collaborative governance as effective systems. These approaches have proven advantageous amidst increasingly complex food supply chains with multiple subjects and sensitive trigger points [4] (pp. 1–16). Since 2013, China has also adopted this governance model as part of its food safety supervision reforms, achieving notable improvements in the quality of food production within supply chains [5,6].

The academic community has extensively researched interdepartmental collaborative governance, focusing on its objectives [7,8], subjects [9,10], and approaches [11,12]. Despite differing interpretations, scholars unanimously agree that interdepartmental collaborative governance involves diverse entities, such as the government, market, and society, all of which interact and participate across traditional organizational boundaries to jointly manage complex social issues [13,14,15]. However, it is important to note that while interdepartmental collaborative governance modes vary, governments bear the primary responsibility for ensuring food safety and designing effective governance systems for it. Therefore, interdepartmental cooperation within governments plays a pivotal and irreplaceable role in enhancing the quality of food production within supply chains [4]. In the Chinese context, Wu et al. reported that within the frameworks of social co-governance and interdepartmental collaborative governance, enterprises are exhibiting an increasing interest in allocating resources for quality improvement. The authors also emphasized the significant value of intergovernmental information exchange in the digital era for enhancing food output quality in supply chains [16]. However, a review of the current literature reveals a lack of studies on how interdepartmental cooperation influences food output quality via information exchange. Therefore, our study sought to address this gap by establishing a three-party game model within China’s current social co-governance framework and conducting a case study on the pork supply chain. This study sought to establish a three-party game model based on the social co-governance framework, using the pork supply chain in China as a case study. Additionally, we examined how interdepartmental collaborative supervision, influenced by the degree of inter-governmental information flow, affects the output quality of the pork supply chain. Through this research, we aim to contribute new insights into the impacts of interdepartmental cooperation on food output quality based on information exchange.

The structure of this paper is organized as follows: Section 1 is the introduction, which mainly describes the purpose of this study. Section 2 discusses the literature review of social co-governance and interdepartmental collaborative governance, which provides support for follow-up research. Then, in Section 3 and Section 4, the research hypotheses are proposed and a three-party game model is established based on the literature review of social consensus and interdepartmental collaborative governance. Section 5 and Section 6 provide the model calculation results, evolutionary path analysis, and stability condition verification analysis of equilibrium points. Section 7 discusses how subject behavior is influenced by the collaborative supervision between government departments based on information sharing. Finally, Section 8 draws the main conclusions, discusses the policy implications of this study according to the conclusions and Chinese context, and points out the shortcomings of this paper. Collectively, our findings contribute significantly to the development of food safety policy in two main ways. First, we developed an evolutionary game model involving three key stakeholders in the pork supply chain: pig farmers, slaughterers, and pork manufacturers. This model also incorporates a social co-governance system that includes government departments, enterprises, and consumers. Second, we focused on the role of information sharing and analyzed how interdepartmental collaboration among government bodies affects the strategic behavior of pig farmers, slaughterers, and pork manufacturers. Particularly, we examined how this collaboration influences the quality of pork and pork products. And meaningful conclusions have been drawn, which are beneficial for deepening collaborative supervision among government departments and improving the output quality of pork and pork products in the supply chain system. Significant conclusions have been reached, which are beneficial for enhancing collaborative supervision among government departments and improving the output quality of pork and pork products in the supply chain system. Moreover, this study draws several meaningful conclusions that can help enhance collaborative supervision among government departments and improve the output quality of pork and pork products in the supply chain system.

## 2. Literature Review on Social Co-Governance and Interdepartmental Cooperative Mechanisms for Food Safety

Zero risk in food safety is unattainable. In reality, food safety risks have been inherent since the dawn of human society and have evolved alongside human and societal development. On a global scale, the evolution of food safety risks is known to be correlated with the economic and technological development of a given country. This relationship is often depicted as an inverted U-shaped curve, known as the food safety risk Kuznets curve (Food Safety Risk Kuznets Curve, FKC), as illustrated in Figure 1 [17].

However, the establishment and improvement of food safety risk governance often lag behind the development of the food industry and societal needs, with governance systems often struggling to keep pace with the evolution and development of food-related risks [18]. One illustrative example is the serious cases of food fraud in the United Kingdom during the early 19th century. Merchants and manufacturers adulterated many food products without regard for consumer health [19]. This problem peaked in the mid-19th century, with the adulteration rate of milk in the London market reaching levels as high as 50% [20]. Despite the establishment of government-led regulatory agencies by this time, the authorities were unable to curb the widespread adulteration in the food industry. In 1860, the UK implemented the Adulteration Act of Food and Drink, along with a series of improved food safety systems. However, it was not until the early 20th century that food fraud and adulteration in the UK began to be preliminarily controlled [21].

From the perspective of the evolution of public management and political science, the reform of social public affairs management systems, including food safety, in Western developed countries can be broadly divided into three stages. The first stage is the reform of the traditional bureaucratic system. Bureaucratic management created a rigid system, and the expansion and overlapping functions of government institutions led to a separation among government, market, and society [22]. To address this issue, Western countries began reforming the traditional bureaucratic system in the 1970s. Public management gradually shifted from a traditional liberal or conservative approach to a limited intervention approach, with governance mechanisms evolving from traditional to holistic [23]. The second stage consists of the implementation of new public management (New Public Management, NPM). In the late 1970s, Western countries adopted NPM to address complex public affairs involving the government and society. However, after more than 20 years of reform, the national governance systems of Western countries have become increasingly fragmented [24,25,26]. Governance entities, including the government, market, and society, each operate independently, which can create new gaps or increase differences between and within these participating entities [24,27]. Consequently, in the mid to late 1990s, Western countries initiated the so-called Post-New Public Management (post-NPM) reform. This reform not only leveraged the roles of government and market mechanisms but also introduced the participation of social forces such as the news media, non-governmental organizations, and consumers, thereby delegating regulatory power to multiple non-governmental entities. Compared to traditional governance methods, social co-governance ensures food safety at a lower cost and with more efficient resource allocation. Therefore, it has become a fundamental approach to managing food safety risks in many countries [28]. The reform also actively promoted collaborative governance between government departments, the public and private sectors, and social sectors. For example, to enhance internal collaboration among government departments, Western countries proposed interdepartmental collaborative governance models such as the whole-of-government and joined-up government models [29,30,31,32]. These models aimed to eliminate traditional government structures and institutional arrangements by integrating and coordinating departments, dismantling inherent barriers, and creating interdepartmental collaborative governance mechanisms among multiple entities. This led to the development of social co-governance and interdepartmental collaborative governance for food safety [33]. Extensive literature and decades of practical exploration in Western developed countries have confirmed the success of social co-governance and interdepartmental collaborative mechanisms. For example, recent studies have demonstrated that the risk level in the food supply chain is negatively correlated with the level of social co-governance and interdepartmental collaboration, meaning that increased social oversight and collaboration significantly reduce food-related risks [34,35]. Therefore, social co-governance and interdepartmental collaboration mechanisms have become mainstream systems for managing food safety risks in many countries, including China [36,37].

The food supply chain is highly complex, involving multiple stages such as farming, processing, manufacturing, storage, logistics, transportation, and consumption. Consequently, governments implement multi-departmental supervision mechanisms tailored to these different stages [6,38]. For instance, the United States established its Federal Food Safety Regulatory Agency in 1906, with 15 government departments participating in food safety supervision at the federal level [39]. Despite the many advantages of multi-departmental collaborative supervision, the relative independence and decentralized focus of government departments often resulted in supervision loopholes and even regulatory failure [40,41,42,43,44]. Prior to 2018, more than 10 government departments in China were responsible for pork safety supervision, yet pork safety incidents occurred frequently. The 2011 Shuanghui clenbuterol incident is a notable example. Due to inadequate supervision by the agriculture and rural departments, farmers extensively used clenbuterol and acquired various certificates through bribery. Industrial and commercial departments failed to inspect pork quality thoroughly, only verifying these false certificates. Consequently, clenbuterol-contaminated pork reached the market [45,46]. In 2014, pigs raised in Gao’an, Jiangxi Province, China, some carrying the foot-and-mouth disease virus (a class-A severe infectious disease), were bought by traders and processed into pork, which was then sold to seven provinces and cities, including Guangdong, Hunan, Chongqing, Henan, Anhui, Jiangsu, and Shandong. The annual sales exceeded 20 million yuan, seriously endangering consumer health. This incident was primarily due to long-term regulatory gaps among the agriculture and rural departments, the industrial and commercial departments, the food supervision department, and the public security department [47]. Food safety incidents are not unique to China. In 2007, over 200 people were hospitalized in Mexico due to poisoning from pork containing clenbuterol [48]. These incidents highlight the crucial importance of interdepartmental government supervision in ensuring food safety.

Interdepartmental collaborative governance among government departments involves multiple interdependent departments overcoming barriers to information and resource sharing by horizontally or vertically crossing government departments at the same or different levels. This approach addresses issues such as unclear powers and responsibilities, mutual isolation, function decentralization, and fragmented supervision among departments and regions at different levels caused by overlapping and ambiguous functions [14,49,50]. Various modes of interdepartmental collaboration exist among government departments, enabling them to adapt to the complexity of multiple links and long chains in the food supply chain and improve the quality of food output. In China, government departments can undertake joint activities and special law enforcement efforts to address prominent food safety issues. They can also develop and implement normative agreements or informal collaboration agreements [15]. Among these regulatory measures, establishing and improving inter-governmental food safety information-sharing mechanisms is one of the most fundamental and cost-effective ways to address fragmented inter-governmental supervision [4]. In the information age, effective information-sharing mechanisms can integrate the power chain, responsibility chain, and information chain among government departments, significantly promoting interdepartmental collaborative governance [51,52].

The coordination degree refers to the synergetic effect achieved through interdepartmental collaboration and interaction between organizations, particularly government departments. It represents the ability of individuals or organizations to produce collaborative effects that exceed the sum of their respective capabilities through joint efforts. A higher coordination degree indicates stronger collaboration and coordination abilities, leading to better governance outcomes [53]. Thus, under an effective information-sharing mechanism, the degree of interdepartmental collaborative governance or supervision among government departments can influence the behavior of the food industry and the quality of food output in the supply chain. In the face of a complex and changeable food supply chain, it is crucial for government departments to implement collaborative supervision according to their functions based on shared information, and enhance collaborative supervision and regulatory awareness, as this will significantly restrict the violations of food enterprises, thereby improving food quality.

## 3. Research Hypothesis

In recent decades, pork has consistently accounted for more than 60% of all meat consumption in China, making it the largest pork-consuming country worldwide. The long-standing habit of pork consumption has led to a stable demand for this product, ensuring a solid demand in the future [54]. It is anticipated that the total demand for pork will reach 64.1138 million tons in 2035 [55]. Figure 2 illustrates the main risks in China’s pork supply chain system, highlighting that the behaviors of pig farmers, pig slaughterers, and pork processing manufacturers are the primary actors affecting the output quality of pork and pork products. These three groups often adopt improper production methods in pursuit of economic benefits.

Therefore, the present study simplifies the entities involved into three main groups: pig farmers (including farms and breeding enterprises), pig slaughterers, and pork processing manufacturers. These groups are referred to as farmers, slaughterers, and manufacturers, and are denoted as *X*, *Y*, and *Z*, respectively (in this study, the behavioral subjects of feed processing, transportation, circulation, and sales are not included in the supply chain system). Within the social co-governance framework, there are additional stakeholders in the supply chain, such as government bodies, consumers, and social organizations (although there are various ways for the government to supervise and penalize *X*, *Y*, and *Z*, our study exclusively focused on two mechanisms: supervision sampling and economic penalties. The social forces include consumers, non-governmental organizations, and news organizations, among other factors, whereas only the social force of consumers is considered in this study. Moreover, this study assumed that the only way for consumers to participate in social co-governance is to report quality and safety problems and that the report on the quality and safety of pork products produced by pork manufacturers contributes to the reputation of pork manufacturers. The mutual supervision among *X*, *Y*, and *Z* adheres to market contracts they have signed. This framework not only supports industrial self-discipline within the supply chain system but also underscores the importance of market forces in social co-governance). Considering the complexity of government supervision in China’s real-world context, we formulated a series of hypotheses regarding regulatory oversight based on current government functions. The behavioral subject *X* is supervised solely by the agricultural and rural department (referred to as Department *N*). The pig slaughterers, represented by *Y*, are not only subject to departmental supervision but also interdepartmental collaborative supervision based on the safety information of pork sold in the market and the annual industrial and commercial inspection information. The manufacturers, represented by *Z*, are under the supervision of both Department *S* and the government development and reform department (referred to as Department *G*), which oversees credit supervision. Additionally, the departments of health, commerce, industry, and information technology also participate in the supervision. However, since this study adopts a three-party game model, only eight combination modes can be set for the choice of behavioral strategies of subjects *X*, *Y*, and *Z* within this framework. Given this limitation, our study focuses on the two most important departments that participate in the supervision of *Y* and *Z*.

In China, there are various collaborative supervision methods among government departments, such as joint law enforcement and coordinated anti-counterfeiting efforts. For the purposes of this study, the sharing of regulatory information is taken as the fundamental method of inter-departmental collaborative supervision. Based on this assumption, our study examines the implementation of collaborative supervision among government departments based on information sharing by adjusting the probability of random supervision and further investigates the influence of collaborative supervision on the behavioral strategies of *X*, *Y*, and *Z*. The regulatory information shared among government departments can be categorized into three types: positive, negative, and neutral. This shared information represents the positive, negative, and normal behaviors of *X*, *Y*, and *Z*, respectively. Moreover, different types of shared information can lead to varying changes in the probability of random inspection by government departments. When the shared information is positive, the government department receiving the information may maintain or appropriately reduce the probability of random inspection for the related subjects. Conversely, when negative information is transmitted, the government department receiving the information will increase the random inspection probability and disclose the behavioral information of related subjects, impacting their market reputation and compelling them to change their risky behavior. When neutral information is transmitted, the government department receiving the information may keep the random inspection probability unchanged. In this study, we assumed that the shared information is negative information about the subject’s risky behaviors. Based on this assumption, our study explores the impact of the degree of coordination of collaborative supervision between government departments on the subjects’ choice of behavioral strategies.

Empirical studies in the field of food safety in China have confirmed that collaborative supervision between government departments can influence the behavior of the food industry, playing a positive role in preventing and controlling food safety risks [4,35,56]. Assume that the coordination degrees (ω) of collaborative supervision between government departments are identical, and −1<ω<1. Then, ω<0 indicates that the information-sharing mechanism among government departments is ineffective, or the probability of random inspection was not adjusted in time according to the transmitted information. In this case, the actions of government departments are independent. ω>0 indicates that the information-sharing mechanism among government departments is fully functional and the probability of random inspections can be adjusted in a timely manner based on the shared information, thereby achieving effective collaborative supervision. In this context, higher ω values correspond to a greater degree of coordination in collaborative supervision between government departments [57,58,59].

In a market economy, although *X*, *Y*, and *Z* possess varying degrees of interest in ensuring pork safety, their behavioral strategies are primarily driven by their own economic interests and are characterized by limited rationality. Therefore, their initial actions may not be optimal, and they might engage in irregular behaviors during production and operation, leading to pork safety risks. According to the current management norms of the Chinese government, *X*, *Y*, and *Z* not only need to inspect their own products according to industry requirements but must also inspect the products of upstream enterprises, as stipulated by market contracts, to ensure the quality of the products purchased from these upstream enterprises. Figure 3 illustrates the current supervision flow chart of the entire pork supply chain in China.

## 4. Parameter Settings and Model Establishment

According to Figure 3, this paper makes the following assumption, and the relevant parameter declaration is made in Table 1.

**Assumption** **1.***The strategy selection set for pig breeding behaviors of* X *is {safe breeding, risky breeding}, with the corresponding probabilities being* x *and* 1−x *, respectively (* 0≤x≤1*). Assume X chooses the safe breeding behavior. The expected revenue for the pigs bred by this behavior is S*_1_*. In this case, the random inspections by Y based on market contracts and by Department N for the pigs bred by X will not affect the expected revenue of Y.*

Assume that *X* chooses the risky breeding behavior and sells the live pigs to *Y* for an expected revenue of *S*_2_. Since the cost of risky breeding behavior is lower than the cost of safe breeding behavior, then *S*_2_ > *S*_1_. In this scenario, *X* no longer performs self-checks on the quality of live pigs. Assume that the probability of random inspection for *X* by Department *N* is *g*_1_. If Department *N* detects risky breeding behavior, it will impose an economic penalty *P*_1_ on *X* according to relevant regulations, namely, *X* must pay an economic liquidated damage of *m*_1_*H*_1_. *Y* is a downstream enterprise relative to *X*. If *Y* chooses the safe slaughtering strategy, it will randomly inspect the quality of pigs bred by *X*. Assume that the probability of random inspection by *Y* is *m*_1_, and that the liquidated damages determined by the mutual contract between *X* and *Y* is set as *H*_1_. If *Y* finds that the pigs bred by *X* are at risk, *X* must pay an economic liquidated damage of *m*_1_*H*_1_ to *Y*. Thus, the expected net revenue of *X* is *S*_2_ − *g*_1_*P*_1_ − *m*_1_*H*_1_. When *Y* chooses to engage in risky slaughtering practices, there is no need for *X* to pay the economically liquidated damages. In this case, the expected net revenue of *X* is *S*_2_ − *g*_1_*P*_1_.

**Assumption** **2.***The strategy selection set for pig slaughtering behaviors of Y is {safe slaughtering, risky slaughtering}, and the corresponding probabilities are*  y *and* 1−y*, respectively (*0≤y≤1*). The pork slaughtered by Y is either sold to Z for manufacturing pork products or sold directly to consumers on the market. As previously mentioned, the collaborative supervision of multiple departments on Y involves only Departments N and S. Department N is responsible for sharing the quality safety information of the pork slaughtered by Y with Department S. Based on the shared information, Department S will adjust the probability of random inspection for the pork slaughtered by Y that is sold on the market. The results of these random inspections are then fed back to Department N, thereby completing the collaborative supervision of Y in the slaughtering process.*

If *Y* chooses the safe slaughtering behavior, its expected revenue is *S*_3_. In this scenario, the consumers’ reports, the safe breeding behavior of *X*, the random inspection of *Z* based on market contracts, and the collaborative supervision implemented by Departments *N* and *S* will not affect its expected revenue. However, if *X* chooses the risky breeding behavior, *Y* can gain the liquidated damages *m*_1_*H*_1_ from *X*, and the expected net revenue of *Y* is *S*_3_ + *m*_1_*H*_1_.

If *Y* chooses to engage in risky slaughtering behavior, its expected revenue is *S*_4_. Since the input cost of risky slaughtering is lower than that of safe slaughtering, *S*_4_ > *S*_3_. In this scenario, *Y* no longer self-checks the quality of slaughtered pork and no longer conducts random inspections on the quality of purchased live pigs bred by *X*.

Assume that within the context of an information-sharing mechanism and collaborative supervision, the probability of random inspection by Department *N* on the pork slaughtered by *Y* is *g*_2_, which varies with the change of ω between Departments *N* and *S*. Moreover, it is assumed that if Department *N* detects risky behavior by *Y* during random inspection, it will impose an economic penalty *P*_2_ on *Y*, meaning that *Y* will face a penalty of *g*_2_*P*_2_ from Department *N*. Similarly, if Department *S* finds that pork slaughtered by *Y* being sold on the market is at risk during a random inspection, it will impose an economic penalty *P*_3_ on *Y* according to related regulations and laws. If the probability of random inspection by Department *S* is *g*_3_, *Y* will be subject to an economic punishment of *g*_3_*P*_3_ from Department *S*. Furthermore, when consumers buy risky pork in the market, there is a probability that they will report *Y*. Assume that the reporting probability is *m*_2_. If Department *S* confirms the report, *Y* must pay a compensation of *H*_2_ to the consumers, that is, an economic compensation of *m*_2_*H*_2_ must be paid by *Y* to the consumers. If *Z* chooses the safe processing strategy, it will randomly inspect the pork purchased from *Y*. Assume that the probability of random inspection is *m*_3_, and the economic liquidated damage determined in the contract signed by *Y* and *Z* is *H*_3_. Then, *Y* must pay a liquidated damage of *m*_3_*H*_3_ to *Z*. In this case, the expected net revenue of *Y* is *S*_4_ − *g*_2_*P*_2_ − *g*_3_*P*_3_ − *m*_2_*H*_2_ − *m*_3_*H*_3_. If *Z* chooses the risky processing strategy, there is no need for *Y* to pay liquidated damages, and the expected net revenue of *Y* is *S*_4_ − *g*_2_*P*_2_ − *g*_3_*P*_3_ − *m*_2_*H*_2_.

**Assumption** **3.***The strategy selection set for pig product processing behaviors of Z is {safe processing, risky processing}, and the corresponding probabilities are*  z *and* 1−z *(*0≤z≤1*), respectively. In this scenario, the processing behaviors of Z are subject to the collaborative supervision of Departments S and G. Department S conducts random inspections on the safety of pork products processed by Z, whereas Department G releases the credit information status of Z through the National Enterprise Credit Information Publicity System. This includes supervision and random inspection information provided by Department S, as well as any records of dishonesty. If the released credit information is negative, the market reputation and economic revenue of Z will be adversely affected.*

If *Z* chooses the safe processing behavior, its expected revenue is *S*_5_. In this case, the behavior of *Z* is not risky, resulting in no consumer reports, and the collaborative supervision of Departments *S* and *G* does not affect the expected revenue of *Z*. When *Y* chooses the risky slaughtering behavior, *Z* can claim liquidated damages of *m*_3_*H*_3_ from *Y*. Thus, the expected net revenue of *Z* is *S*_5_ + *m*_3_*H*_3_.

If *Z* chooses the risky processing behavior, it neither self-checks the processed pork products nor conducts random inspections on the quality of pork slaughtered by *Y*. Assume the expected revenue of *Z* in this case is *S*_6_, where *S*_6_ > *S*_5_. Within the context of information sharing and collaborative supervision, the probability of random inspection on *Z* by Department *S* is *g*_4_. If Department *S* finds that *Z* is engaging in risky behavior, it will impose an economic penalty *P*_4_ on *Z* according to related regulations and laws, with *Z* incurring a penalty of *g*_4_*P*_4_ from Department *S*. If consumers purchase risky pork products in the market, they may report *Z* to Department *S*. Assume the probability of consumers reporting *Z* to Department *S* is *m*_4_. If Department *S* confirms the consumer report, *Z* must pay economic compensation *H*_4_ to the consumers in accordance with the regulations. That is, *Z* must pay economic compensation *m*_4_*H*_4_. Additionally, the negative market reputation will cause an expected economic loss *H*_5_ for *Z*. Assuming that the probability of consumers being aware of the negative credit information released by Department *G* is *m*_5_, and then *Z* will bear an economic loss of *m*_5_*H*_5_ due to the negative market reputation. Thus, the expected net revenue of *Z* is *S*_6_ − *g*_4_*P*_4_ − *m*_4_*H*_4_ − *m*_5_*H*_5_. 

According to the above research assumptions and the game relationships among *X*, *Y*, and *Z*, the payoff matrix of the trilateral game can be obtained, as shown in Table 2.

## 5. Model Computation and Evolutionary Path Analysis

According to the payoff matrix in Table 2, the replicator dynamic equations of *X*, *Y*, and *Z* can be solved, respectively. The stability strategy for each subject can then be analyzed using the stability theory of differential equations.

### 5.1. Selection Strategy for Behaviors of Three Subjects

Assume *E*_11_, *E*_12_, and $\bar{E_{1}}$ represent the expected profits of safe breeding, risky breeding, and the average expected profit for *X*, respectively. Assume the replicator dynamic equation corresponding to the selection of strategy for the pig breeding behavior of *X* is F(x). Then, according to Table 2, the following expressions (Equations (1)–(3)) and the replicator dynamic sub-equation of *X* (Equation (4)) can be obtained.
(1)$$\begin{array}{c} E_{11}=yz(S_{1})+(1-y)z(S_{1})+y(1-z)(S_{1})\\ +(1-y)(1-z)(S_{1}) \end{array}$$
(2)$$\begin{array}{ll} E_{12} & =yz(S_{2}-g_{1}P_{1}-m_{1}H_{1})+(1-y)z(S_{2}-g_{1}P_{1})\\ & +y(1-z)(S_{2}-g_{1}P_{1}-m_{1}H_{1})+(1-y)(1-z)(S_{2}-g_{1}P_{1}) \end{array}$$
(3)$$\bar{E_{1}}=xE_{11}+(1-x)E_{12}$$
(4)$$\begin{array}{c} F(x)=\frac{dx}{dt}=x(E_{11}-\bar{E})=x(1-x)(E_{11}-E_{12})\\ =x(1-x)[S_{1}-S_{2}+g_{1}P_{1}+ym_{1}H_{1}] \end{array}$$

Assume *E*_21_, *E*_22_, and $\bar{E_{2}}$ represent the expected profits of safe breeding, risky breeding, and the average expected profit for *Y*, respectively. Assume the replicator dynamic equation corresponding to the selection of strategy for the pig slaughtering behavior of *Y* is F(y). Then, according to Table 2, the following expressions (Equations (5)–(7)) and the replicator dynamic sub-equation of *Y* (Equation (8)) can be obtained.
(5)$$\begin{array}{ll} E_{21} & =xz(S_{3})+(1-x)z(S_{3}+m_{1}H_{1})+x(1-z)(S_{3})\\ & +(1-x)(1-z)(S_{3}+m_{1}H_{1}) \end{array}$$
(6)$$\begin{array}{ll} E_{22} & =xz(S_{4}-g_{2}P_{2}-g_{3}P_{3}-m_{2}H_{2}-m_{3}H_{3})\\ & +(1-x)z(S_{4}-g_{2}P_{2}-g_{3}P_{3}-m_{2}H_{2}-m_{3}H_{3})\\ & +x(1-z)(S_{4}-g_{2}P_{2}-g_{3}P_{3}-m_{2}H_{2})\\ & +(1-x)(1-z)(S_{4}-g_{2}P_{2}-g_{3}P_{3}-m_{2}H_{2}) \end{array}$$
(7)$$\bar{E_{2}}=yE_{21}+(1-y)E_{22}$$
(8)$$\begin{array}{ll} F(y) & =\frac{dy}{dt}=y(E_{21}-\bar{E})=y(1-y)(E_{21}-E_{22})\\ & =y(1-y)[S_{3}-S_{4}+g_{2}P_{2}+g_{3}P_{3}+(1-x)m_{1}H_{1}+m_{2}H_{2}+zm_{3}H_{3}] \end{array}$$

Assume *E*_31_, *E*_32_, and $\bar{E_{3}}$ represent the expected profits of safe breeding, risky breeding, and the average expected profit for *Z*, respectively. Assume the replicator dynamic equation corresponding to the selection of strategy for the pork processing behavior of *Z* is F(z). Then, according to Table 2, the following expressions (Equations (9)–(11)) and the replicator dynamic sub-equation of *Z* (Equation (12)) can be obtained.
(9)$$\begin{array}{ll} E_{31} & =xy(S_{5})+(1-x)y(S_{5})+x(1-y)(S_{5}+m_{3}H_{3})\\ & +(1-x)(1-y)(S_{5}+m_{3}H_{3}) \end{array}$$
(10)$$\begin{array}{ll} E_{32} & =xy(S_{6}-g_{4}P_{4}-m_{4}H_{4}-m_{5}H_{5})\\ & +(1-x)y(S_{6}-g_{4}P_{4}-m_{4}H_{4}-m_{5}H_{5})\\ & +x(1-y)(S_{6}-g_{4}P_{4}-m_{4}H_{4}-m_{5}H_{5})\\ & +(1-x)(1-y)(S_{6}-g_{4}P_{4}-m_{4}H_{4}-m_{5}H_{5}) \end{array}$$
(11)$$\bar{E_{3}}=zE_{31}+(1-z)E_{32}$$
(12)$$\begin{array}{ll} F(z) & =\frac{dz}{dt}=z(E_{31}-\bar{E})=z(1-z)(E_{31}-E_{32})\\ & =z(1-z)[S_{5}-S_{6}+g_{4}P_{4}+(1-y)m_{3}H_{3}+m_{4}H_{4}+m_{5}H_{5}] \end{array}$$

According to the above three replicator dynamic sub-equations (Equations (4), (8) and (12)), the tripartite replicator dynamic equation set can be obtained as Equation (13).
(13)$$\left\{ \begin{array}{l} F(x)=x(1-x)[S_{1}-S_{2}+g_{1}P_{1}+ym_{1}H_{1}]\\ F(y)=y(1-y)[S_{3}-S_{4}+g_{2}P_{2}+g_{3}P_{3}+(1-x)m_{1}H_{1}+m_{2}H_{2}+zm_{3}H_{3}]\\ F(z)=z(1-z)[S_{5}-S_{6}+g_{4}P_{4}+(1-y)m_{3}H_{3}+m_{4}H_{4}+m_{5}H_{5}] \end{array}\right.$$

### 5.2. Evolutionary Paths of Three Subjects

(1)Pig farmers

According to the stability theory of differential equations, the following conditions must be satisfied when the probability of *X* safely breeding pigs is maintained in a stable state: F(x)=0 and F’(x)=d(F(x))/dx<0.

When $y=y*=\frac{S_{2}-S_{1}-g_{1}P_{1}}{m_{1}H_{1}}$, F’(x)=0 and all values of x are in the stable state of behavioral strategy evolution. When y>y*, F’(1)<0 and x=1 is the evolutionary stable strategy, meaning that *X* chooses safe breeding strategies for live pigs. In contrast, when y<y*, F’(0)<0 and x=0 is the evolutionary stable strategy, which means that *X* chooses the behavioral strategy of risky breeding for live pigs. The phase diagrams of the strategy evolution of *X* breeding live pigs are shown in Figure 4, in which the arrow tips represent the evolution of x to x=0 or x=1.

(2)Pig slaughterers

Similarly, the following conditions must be satisfied when the probability of *Y* choosing safe slaughtering behavior is maintained in a stable state: F(y)=0 and F’(y)=d(F(y))/dy<0.

When $z=z*=\frac{S_{4}-S_{3}-g_{2}P_{2}-g_{3}P_{3}+(x-1)m_{1}H_{1}-m_{2}H_{2}}{m_{3}H_{3}}$, F’(y)=0 and all values of y are in the stable state of behavioral strategy evolution. When z>z*, F’(1)<0, and y=1 is the evolutionary stable strategy, which means that *Y* chooses the behavioral strategy of safe slaughtering for live pigs. In contrast, when z<z*, F’(0)<0, and y=0 is the evolutionary stable strategy, meaning that *Y* chooses the behavioral strategy of risky slaughtering of live pigs. The phase diagrams of the strategy evolution of *Y* slaughtering live pigs are shown in Figure 5, in which the arrow tips represent the evolution of y to y=0 or y=1.

(3)Pork processing manufacturers

The following conditions must be satisfied when the probability of *Z* choosing safe pork processing behavior is maintained in a stable state: F(z)=0 and F’(z)=d(F(z))/dz<0.

When $y=y*=\frac{S_{5}-S_{6}+g_{4}P_{4}+m_{4}H_{4}+m_{5}H_{5}}{m_{3}H_{3}}+1$, F’(z)=0 and all values of z are in the stable state of behavioral strategy evolution. When y>y*, F’(1)<0 and z=1 is the evolutionary stable strategy, meaning that *Z* chooses the behavioral strategy of safe pork processing. In contrast, when y<y*, F’(0)<0, and z=0 is the evolutionary stable strategy, meaning that *Z* chooses the behavioral strategy of risky pork processing. The phase diagrams of the strategy evolution of *Z* processing pork behaviors are shown in Figure 6, in which the arrow tips represent the evolution of z to z=0 or z=1.

### 5.3. Stability Analysis of Equilibrium Point of Evolutionary Game

In the replicator dynamic Equation (13), let $\frac{dx}{dt}=0$, $\frac{dy}{dt}=0$, and $\frac{dz}{dt}=0$. Then, in the three-dimensional space, there are eight equilibrium points on V={(x,y,z)|0≤x≤1,0≤y≤1,0≤z≤1}: *M*_1_(1,1,1), *M*_2_(0,1,1), *M*_3_(1,0,1), *M*_4_(1,1,0), *M*_5_(0,0,1), *M*_6_(0,1,0), *M*_7_(1,0,0), and *M*_8_(0,0,0).

To analyze the asymptotic stability of equilibrium points, the Jacobian matrix and its characteristic value (*λ*) can be calculated according to Equation (14). According to Lyapunov stability theory [60], when all the characteristic values of the Jacobian matrix satisfy the condition of *λ* < 0, the equilibrium points are asymptotically stable. The calculation results are provided in Table 3.
(14)$$\begin{array}{c} K=\left[\begin{array}{ccc} \frac{\partial F(x)}{\partial x} & \frac{\partial F(x)}{\partial y} & \frac{\partial F(x)}{\partial z}\\ \frac{\partial F(y)}{\partial x} & \frac{\partial F(y)}{\partial y} & \frac{\partial F(y)}{\partial z}\\ \frac{\partial F(z)}{\partial x} & \frac{\partial F(z)}{\partial y} & \frac{\partial F(z)}{\partial z} \end{array}\right]=\left[\begin{array}{ccc} a11 & a12 & a13\\ a21 & a22 & a23\\ a31 & a32 & a33 \end{array}\right]=\\ \left[\begin{array}{ccc} (1-2x)\left(\begin{array}{c} S_{1}-S_{2}\\ +g_{1}P_{1}+ym_{1}H_{1} \end{array}\right) & x(1-x)m_{1}H_{1} & 0\\ y(y-1)m_{1}H_{1} & (1-2y)\left[\begin{array}{c} S_{3}-S_{4}+g_{2}P_{2}+g_{3}P_{3}\\ +(1-x)m_{1}H_{1}+m_{2}H_{2}+zm_{3}H_{3} \end{array}\right] & y(1-y)m_{3}H_{3}\\ 0 & z(z-1)m_{3}H_{3} & (1-2z)\left[\begin{array}{c} S_{5}-S_{6}+g_{4}P_{4}\\ +(1-y)m_{3}H_{3}+m_{4}H_{4}+m_{5}H_{5} \end{array}\right] \end{array}\right] \end{array}$$

## 6. Verification Analysis of Stability Conditions of Equilibrium Points

To verify the stability conditions of each equilibrium point listed in Table 3, a numerical simulation of the specific evolutionary paths of the trilateral game of *X*, *Y*, and *Z* was conducted by using the Matlab software R2023a. Based on the number of subjects engaging in risky behaviors in the pork supply chain, the eight equilibrium points obtained above can be classified into four types of scenarios: zero-subject risky behavior, single-subject risky behavior, double-subject risky behavior, and tri-subject risky behavior. These are referred to as Type I, Type II, Type III, and Type IV scenarios, respectively. *M*_1_(1,1,1) represents a scenario in which all three subjects exhibit no risky behaviors, which falls under the zero-subject risky behavior category (Type I); *M*_2_(0,1,1), *M*_3_(1,0,1), and *M*_4_(1,1,0) represent scenarios in which one of the three subjects engages in risky behaviors, falling under the single-subject risky behavior category (Type II); *M*_5_(0,0,1), *M*_6_(0,1,0), and *M*_7_(1,0,0) represent scenarios in which two of the three subjects engage in risky behaviors, falling under the double-subject risky behavior category (Type III); *M*_8_(0,0,0) represents a scenario in which all three subjects engage in risky behaviors, falling under the tri-subject risky behavior category (Type IV). According to the research hypothesis and Table 3, the parameter values for each equilibrium point were established as shown in Table 4, after which numerical simulations were conducted to analyze their stability.

### 6.1. Type I: Zero-Subject Risky Behavior Scenario

Since all the parameters satisfy condition (1) in Table 3, the initial intentions of *X*, *Y*, and *Z* can be set as (0.3,0.4,0.9), (0.5,0.6,0.7), and (0.2,0.5,0.8), respectively. Using these initial intentions, the selection of their behavioral strategies under different initial conditions can be simulated. The simulation results in Figure 7 show that under the three initial intentions, the behavioral strategies of *X*, *Y*, and *Z* eventually evolve into *M*_1_(1,1,1), representing the state of {safe breeding, safe slaughtering, and safe processing}. This indicates that the evolution of the system to the Type I situation is independent of their respective initial intentions. However, the initial intention only affects the time required for the system to evolve to the equilibrium point. The time required for the system to reach a stable state under higher initial intentions is shorter than that under lower initial intentions. In fact, when the economic revenue of *X*, *Y*, and *Z* choosing risky behavioral strategies is greater than that of choosing safe behavioral strategies, they all have an initial motivation to choose risky behavioral strategies. However, in the Condition (1) scenario, $g_{1}P_{1}+m_{1}H_{1}>S_{2}-S_{1}$, $g_{2}P_{2}+g_{3}P_{3}+m_{2}H_{2}+m_{3}H_{3}>S_{4}-S_{3}$, and $g_{4}P_{4}+m_{4}H_{4}+m_{5}H_{5}>S_{6}-S_{5}$. In other words, when the total economic losses borne by *X*, *Y*, or *Z* exceed the additional economic gains from their choice of risky behavior strategies, and the total economic losses caused by negative market reputation exceeds the additional economic gains of *X*, *Y*, and *Z* resulting from choosing risky behavior strategies, they will choose safe behavioral strategies regardless of their initial intentions. The equilibrium point will eventually evolve to *M*_1_(1,1,1), and the system will eventually evolve to a Type I scenario. In this case, pork safety information flows effectively among Departments *N*, *S*, and *G*. Department *G* releases the credit information on time, and the coordination degree of collaborative supervision among government departments is high. Furthermore, enterprises are mutually restricted based on market mechanisms, and consumers actively participate in reporting risky behaviors. Under these circumstances, the social co-governance system is at a high level, and the overall risk of the supply chain is minimized.

### 6.2. Type II: Single-Subject Risky Behavior Scenario

Given that all parameters satisfy conditions (2), (3), and (4) in Table 3, the initial intentions of *X*, *Y*, and *Z* can be set as (0.3,0.4,0.9), (0.5,0.6,0.7), and (0.2,0.5,0.8), respectively. These values are then used to conduct a simulation similar to that in situation I. The simulation results shown in Figure 8, Figure 9 and Figure 10 indicate that, under these initial intentions, the behavioral strategies of *X*, *Y*, and *Z* finally evolve into {risky breeding, safe slaughtering, safe processing}, {safe breeding, risky slaughtering, safe processing}, and {safe breeding, safe slaughtering, risky processing}, respectively. Similarly, the evolution of the system into a Type II scenario is determined by their stability conditions and not their initial intentions. The evolution rate of the behavioral strategy with the initial intention is similar to that in the Type I scenario, which will not be detailed here. Take *M*_4_(1,1,0) as an example. Under condition (4), that is, $g_{4}P_{4}+m_{4}H_{4}+m_{5}H_{5}<S_{6}-S_{5}$, *Z* chooses the risky behavior strategy. In this case, the sum of the economic penalties imposed by Department *S* according to the pertinent laws and regulations, the compensation paid to consumers in accordance with regulations, and the total economic loss caused by negative market reputation is lower than the additional economic gains that can be obtained by risky behaviors. Consequently, the joint supervision of the government, consumers, and the market fails to form an effective deterrent, leading to relaxed random inspections of *Y* by *Z*. Notably, at this point, $g_{2}P_{2}+g_{3}P_{3}+m_{2}H_{2}>S_{4}-S_{3}$, meaning that Departments *S* and *N* will share the risky behavior information of *Z* and increase the probability of random inspection on *Y* to curb their motivation for risky slaughtering. Meanwhile, consumers will actively report the purchased risky pork. Under these circumstances, only *Z* chooses the risky behavioral strategy of risky processing. Thus, the equilibrium point will eventually evolve to *M*_4_, and the system will eventually evolve to a Type II scenario, with the overall risk level at a medium level. In this situation, since *Z* chooses the risky processing strategy and loosens the random inspection on *Y*, the mutual supervision between upstream and downstream enterprises is insufficient, making it difficult for the market mechanism to fully function. Overall, in the single-subject risky behavior scenario, the social co-governance system is at a medium level.

### 6.3. Types III and IV: Double-Subject or Tri-Subject Risky Behavior Scenarios

Similarly, since all the parameters satisfy conditions (5–8) in Table 3, the initial intentions of *M*_5_(0,0,1), *M*_6_(0,1,0), and *M*_7_(1,0,0) can be set as (0.2,0.4,0.9), (0.5,0.6,0.7), and (0.6,0.7,0.8), respectively, whereas those of *M*_8_(0,0,0) can be set as (0.3,0.4,0.9), (0.5,0.6,0.7), and (0.2,0.5,0.8), respectively. These settings of initial intentions are then used to conduct the same simulation as for situations I and II, and the simulation results are shown in Figure 11, Figure 12, Figure 13 and Figure 14. Under different initial intentions, the behavioral strategies of *X*, *Y*, and *Z* finally evolve into {risky breeding, risky slaughtering, safe processing}, {risky breeding, safe slaughtering, risky processing}, {safe breeding, risky slaughtering, risky processing}, and {risky breeding, risky slaughtering, risky processing}, respectively. The evolution rates of their behavioral strategies are similar to those in the Type I and II scenarios. Once again, taking *M*_4_(1,1,0) as an example, after *Z* chooses the risky processing behavior and loosens the random inspection on *Y*, if the degree of information sharing between Departments *N* and *S* is so low that they do not effectively adjust their respective probability of random inspection on *Y*, then $g_{2}P_{2}+g_{3}P_{3}+m_{2}H_{2}<S_{4}-S_{3}$. In other words, the additional economic benefit obtained by *Y* is higher than the sum of the economic penalties from the government and the total compensation paid to consumers for reporting. In this case, *Y* will choose the risky behavior strategy, causing the equilibrium point to evolve to *M*_7_(1,0,0). Consequently, the behavioral strategy set evolves to {safe breeding, risky slaughtering, risky processing}, and the system gradually evolves into a Type III scenario. In scenario III, since *Y* chooses the risky slaughtering behavior and no longer conducts random inspection on *X*, if Department *N* does not increase the probability of random inspection on *X*, that is, $g_{1}P_{1}<S_{2}-S_{1}$, the equilibrium point will evolve to *M*_8_(0,0,0), and their behavioral strategy set will further evolve into {risky breeding, risky slaughtering, risky processing}. Thus, the system will evolve into a type IV scenario, where the overall risk level is the highest. In this case, due to the failure of the information-sharing mechanism among government departments and the low level of collaborative supervision, the degree of mutual restraint between enterprises based on the market mechanism is low, leading to an increased risk in the supply chain. Under these circumstances, the social co-governance system is at its lowest level.

### 6.4. Mutual Relation between Equilibrium Points and Social Co-Governance

There are various forms of risks in the pork supply chain under social co-governance. In this paper, the subjects of social co-governance include the government departments *N*, *S*, and *G*; the market subjects *X*, *Y*, and *Z*; and the consumers. The eight equilibrium points *M*_1_(1,1,1), *M*_2_(0,1,1), *M*_3_(1,0,1), *M*_4_(1,1,0), *M*_5_(0,0,1), *M*_6_(0,1,0), *M*_7_(1,0,0), and *M*_8_(0,0,0) represent the states of the social co-governance system. Based on the research, the relationship between these equilibrium points and the social co-governance level can be illustrated in Figure 15. As seen in the figure, the continuous evolution of the system from *M*_8_(0,0,0) to *M*_1_(1,1,1) reflects a process of gradual adjustment in the behavioral strategies of *X*, *Y*, and *Z*. This evolution is also characterized by continuous improvement in inter-departmental information sharing and collaborative supervision, marking a gradual transition from a low to a high level of co-governance involving all participants, including supply chain subjects, government departments, and consumers.

## 7. Influences of Inter-Departmental Collaborative Supervision Based on Information Sharing on the Selection of Behavioral Strategies

In the previous sections, we discussed and verified the stability conditions of the equilibrium points in Type I, II, III, and IV scenarios. Our findings demonstrated that the system’s evolution to a stable state is related to the social co-governance level of the pork supply chain and is independent of initial intentions. *M*_8_(0,0,0) is the equilibrium point when *X*, *Y*, and *Z* in the pork supply chain system collectively choose the risky behavioral strategy, resulting in the highest risk level for the system. In this section, we focus on this scenario to study the influences of inter-governmental collaborative supervision based on information sharing on the behavioral strategy choices of the involved subjects. Given that *Y* is the most critical subject in the supply chain system, we will emphasize analyzing the effects of collaborative supervision by Departments *N* and *S* on the behavior of *Y*.

### 7.1. Relationship between the Types of Information Transmitted between Government Departments and the Probability of Random Inspection

As discussed above, the information shared between government departments falls into three main categories: positive, negative, and neutral. These categories reflect the different strategies of behavior for *X*, *Y*, and *Z*. It is assumed that Departments *N* and *S* will adjust their random inspection probabilities based on the type of information shared to enhance collaborative supervision. For further analysis, as illustrated in Figure 16-A, an increase in the degree of information sharing between Departments *N* and *S* from *T*_1_ to *T*_2_ results in a shift from negative to positive collaboration for ω. In Figure 16-B, when positive behavior information of *Y* is shared between Departments *N* and *S*, the probability of random inspection from Departments *N* and *S* can remain within an *R*_1_ level or even decrease from *Q*_1_ to *Q*_2_ with the continuous increase in ω. In Figure 16-C, when neutral behavior information is shared between Departments *N* and *S*, the probability of random inspection of Departments *N* and *S* can be maintained at *R*_2_ and *Q*_3_.

In Figure 16-D, the transmission of negative behavior information about *Y* between Departments *N* and *S* indicates *Y*’s adoption of risky slaughtering behavior. This scenario presents two possibilities. First, if the information sharing mechanism between Departments *N* and *S* fails, they will not adjust their respective probability of random inspection on *Y* to implement collaborative supervision, resulting in a negative collaboration state between Departments *N* and *S* (−1<ω<0). Second, if the information sharing mechanism functions effectively, Departments *N* and *S* play an adequate role, Departments *N* and *S* will adjust their inspection probabilities based on the risky behavior state of *Y*. This leads to positive collaboration between Departments *N* and *S* on *Y* (0<ω<1). However, if the value of ω is low, the random inspection probabilities of Departments *N* and *S* will gradually increase from levels *R*_3_ and *Q*_4_ to *R*_4_ and *Q*_5_, respectively. If ω continues to rise until it reaches the highest value of 1, the random inspection probability of Departments *N* and *S* will greatly increase from *R*_4_ and *Q*_5_ to *R*_5_ and *Q*_6_, respectively, thus compelling *Y* to reconsider its choice of slaughtering strategy.

### 7.2. Influences of Changes in Random Inspection Probability of Government Departments on the Behavioral Strategy Choice of Subjects 

Assuming that all other parameters remain unchanged, the probabilities of random inspections by Department *N* on pork slaughtered by *Y* and by Department *S* on pork entering the market are set as *g*_2_ = 0.1, 0.3, 0.6, 0.8 and *g*_3_ = 0.2, 0.4, 0.7, 0.9, respectively. Based on these settings, the evolutionary paths depicted in Figure 17 are obtained.

Once again, the supervision of *Y* will be taken as an example. When the information-sharing mechanism between Departments *N* and *S* fails, they enter a negative collaboration state. In this scenario, they independently conduct random inspections based on predetermined probabilities. Conversely, in a positive collaboration state, assuming *g*_2_ = 0.1 and *g*_3_ = 0.2, the low probabilities of random inspections from Departments *N* and *S*, expressed as $g_{2}P_{2}+g_{3}P_{3}+m_{2}H_{2}+m_{3}H_{3}<S_{4}-S_{3}$, lead *Y* to opt for riskier behavioral strategies to maximize economic gains. As shown in Figure 16-D, as the probabilities of random inspection by Departments *N* and *S* increase from *R*_3_ and *Q*_4_ to *R*_4_ and *Q*_5_, respectively (i.e., *g*_2_ = 0.3 and *g*_3_ = 0.4), *Y* continues to favor riskier strategies, albeit with a slightly slower evolution rate towards risky behavior. With continued improvement in information sharing, the probabilities of random inspection by Departments *N* and *S* increase from *R*_4_ and *Q*_5_ to *R*_5_ and *Q*_6_, respectively (i.e., *g*_2_ = 0.6 and *g*_3_ = 0.7). At this moment, $g_{2}P_{2}+g_{3}P_{3}+m_{2}H_{2}+m_{3}H_{3}>S_{4}-S_{3}$, and the equilibrium point gradually evolves from *M*_8_(0,0,0) to *M*_6_(0,1,0). As the degree of information transmission and sharing reaches an optimal state, where ω tends to 1, the probabilities of random inspection by Departments *N* and *S* increase, respectively, to *g*_2_ = 0.8 and *g*_3_ = 0.9; *Y* adopts safer behavioral strategies, significantly accelerating its evolution towards safer practices.

The behavioral strategies of *X* and *Z* are governed by similar mechanisms to those of *Y*, where an improved ω value due to effective information sharing encourages governmental agencies to enhance regulatory effectiveness, optimize inspection probabilities, and promptly disclose credit information to relevant entities. This ultimately compels *X* and *Z* to adopt safer behavioral strategies. Consequently, the equilibrium point for *X*, *Y*, and *Z* in the system eventually transitions from *M*_8_(0,0,0) to *M*_1_(1,1,1), signifying the attainment of an optimal level of social co-governance within the system.

### 7.3. Influences of the Severity of Economic Penalties from Government Departments on the Selection of Behavioral Strategies

The evolution of behavioral strategy choices among *X*, *Y*, and *Z* discussed earlier is closely linked to the degree of coordination ω in collaborative supervision between government departments. This coordination is highly influenced by changes in the random inspection probabilities of Departments *N* and *S*, as well as the transparency and timeliness of credit information released by Department G. However, the evolution of the behavioral strategies of *X*, *Y*, and *Z* largely depends on the change of economic punishment intensity caused by the changes in random inspection probability and released credit information of government departments, namely, it depends on their respective economic benefits.

Taking the choice of behavioral strategy of *Y* as an example, assuming that other parameters remain unchanged, when Departments *N* and *S* find that *Y* is engaging in risky behaviors during random inspection, they will respectively implement economic penalties *P*_2_ = 2, 4, 6, 8 and *P*_3_ = 3, 5, 7, 9 on *Y*. The evolutionary trajectory of *Y*’s behavioral strategy is depicted in Figure 18. Notably, when *P*_2_ = 2 and *P*_3_ = 3, *Y* opts for risky behavior. As penalties increase to *P*_2_ = 4 and *P*_3_ = 5, *Y* still chooses risky behaviors, albeit at a slower rate. However, as economic penalties escalate further to *P*_2_ = 6 and *P*_3_ = 7, risky behavior transitions into safe behavior, gradually shifting the system’s equilibrium point from *M*_8_(0,0,0) to *M*_6_(0,1,0). When penalties reach *P*_2_ = 8 and *P*_3_ = 9, *Y* consistently adopts safe behavioral strategies, with an accelerated evolution rate. These findings reveal that the increase in economic penalties by Departments *N* and *S* effectively deter *Y* from engaging in risky behaviors. Similarly, changes in economic penalties or alterations in market reputation due to released credit information can also influence the behavioral strategies of *X* and *Z*.

## 8. Conclusions and Policy Implications

In the context of Chinese social co-governance, this paper develops an evolutionary three-party game model involving farmers, slaughterers, and manufacturers. It examines equilibrium stability through numerical simulations and explores how inter-departmental collaborative supervision, facilitated by information-sharing mechanisms, influences the behavioral strategies of these entities. The main conclusions of this study are as follows:

The behavioral strategies of farmers, slaughterers, and manufacturers are influenced by their economic benefits before and after strategy changes. The risk state of the pork supply chain system is categorized into four types (Type I to Type IV) based on changes in behavioral strategies. The degree of risk depends on the social co-governance level, including the coordination of collaborative supervision between government departments, market subjects’ mutual supervision through contracts, consumer reporting intensity, and the collaborative governance capabilities of government, market, and consumers. This finding is consistent with the conclusions of Wu et al. (2024) [16]. As the co-governance level improves, fewer subjects opt for risky behaviors, and vice versa. The system’s final stable state is determined by the co-governance level rather than the initial intentions of the subjects, although initial intentions influence the time it takes to reach stability. This is consistent with the conclusions of Gao et al. (2023) [35].

The behavioral strategies of supply chain subjects change not only with the supervision intensity of individual government departments but also with the coordination degree of collaborative supervision between departments. This is consistent with the conclusions of Li (2024) [61]. A single department’s supervision intensity is influenced by the probability of random inspections and the severity of economic penalties. However, the financial constraints limit inspection probabilities, and penalties must align with the risk level of behaviors. Therefore, the economic penalties of a single department have a limited impact on a subject’s economic benefits in situations involving risky behaviors. This is consistent with the conclusions of Ji (2012) [62]. Effective change in behavioral strategies requires collective action: sharing risky behavior information among departments, jointly increasing inspection probabilities and penalties, transparently releasing credit information by the development and reform department to influence market reputations, and actively addressing consumer reports through compensatory measures ordered by relevant authorities. This conclusion is consistent with those of Zhao et al. (2020) [63].

The research findings outlined above hold significant practical implications for enhancing the supervision of pork and its products’ quality and safety. The behavioral strategies of pork producers and sellers depend primarily on their economic incentives. While these enterprises may aspire to improve product quality and safety, rational behaviors are not naturally incentivized and require government oversight, mutual supervision within the supply chain, and involvement from societal forces such as consumer reporting. In the Chinese context, governmental responsibility outweighs mutual enterprise supervision and consumer reporting in driving these improvements. Elevating the quality standards of pork products in the supply chain depends not only on the legal oversight of individual government departments but also on collaborative governance among the agriculture and rural department, market supervision department, and development and reform department. Our findings demonstrate that effective information-sharing mechanisms between government departments, coupled with timely adjustments to random inspection probabilities and transparent release of market credit information in response to changes in enterprise behaviors, can compel enterprises to adopt safer practices. Thus, establishing robust information-sharing mechanisms represents the most cost-effective approach to implementing social co-governance among government departments, crucially enhancing the quality and safety of pork and pork products in the supply chain. 

In summary, the main contributions of this study are as follows: (1) based on the social co-governance framework, this study established an evolutionary game model among pig farmers, pig slaughterers, and pork processing manufacturers; (2) by focusing on the interdepartmental information sharing mechanism, this study investigated the impact of promoting cooperative governance based on inter-governmental information flow on the behaviors of pig farmers, pig slaughterers, and pork processing manufacturers. For the Chinese government, enhancing the interdepartmental information sharing mechanism through institutional innovation and mechanism reform is crucial. This reform measure not only incurs minimal costs but also aligns with the requirements of social governance during the information age, ultimately contributing to the improvement of pork output quality.

However, our study has some noteworthy limitations. The three-party game model simplifies the pork supply chain to include only pig farmers, pig slaughterers, and pork manufacturers, excluding other vital components such as storage, transportation, sales subjects, and non-governmental organizations or media influences. Additionally, while certain links in the pork supply chain may involve multiple government departments, the study primarily focuses on the two most pivotal departments. Therefore, future research should explore collaborative supervision involving a broader array of departments and consider the positive aspects of information-sharing mechanisms beyond negative impacts and changes in government oversight behaviors.

Despite these limitations, our study’s conclusions underscore the need for government departments to enhance information-sharing mechanisms and enact timely legal measures responsive to enterprise behaviors. This approach enhances targeted supervision of high-risk segments, optimizes resource allocation, and improves overall supervisory efficiency.

## Figures and Tables

**Figure 1 foods-13-02387-f001:**
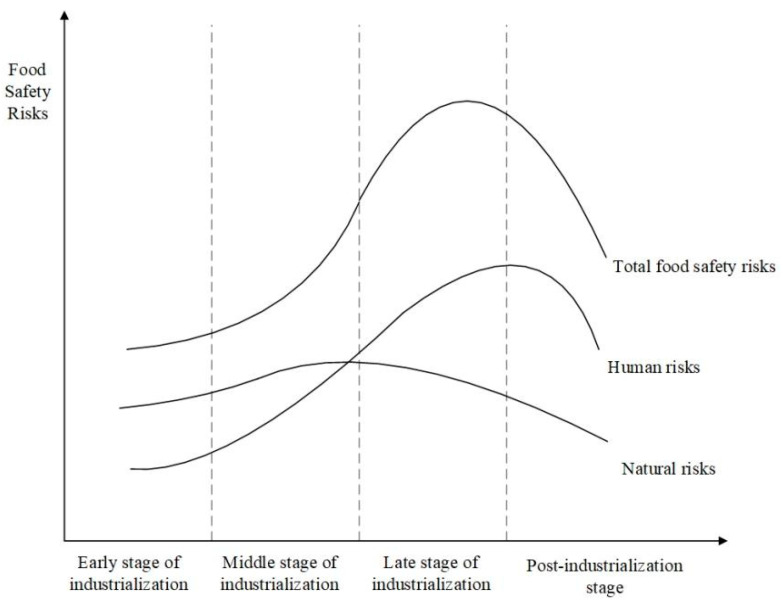
Inverted U-shaped curve relationship between food safety risks and economic development stages.

**Figure 2 foods-13-02387-f002:**
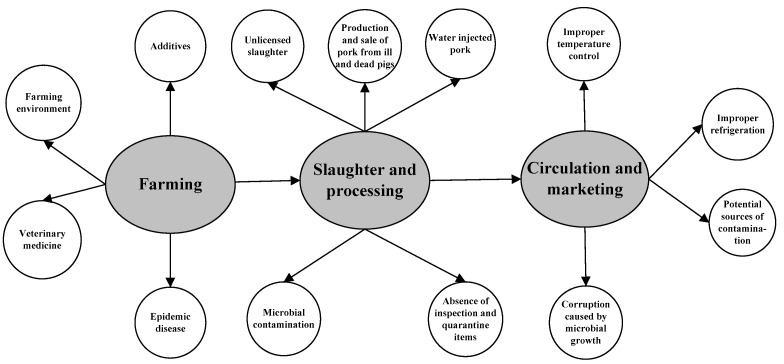
Specific safety risks in the major processes of the pork supply chain.

**Figure 3 foods-13-02387-f003:**
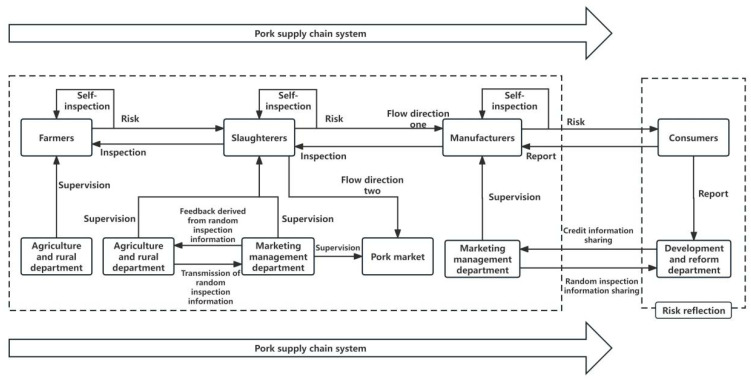
Supervision flow chart of the entire pork supply chain in China.

**Figure 4 foods-13-02387-f004:**
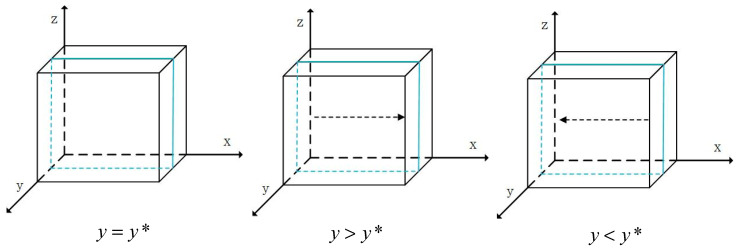
Phase diagrams of the evolution of the breeding behaviors of pig farmers.

**Figure 5 foods-13-02387-f005:**
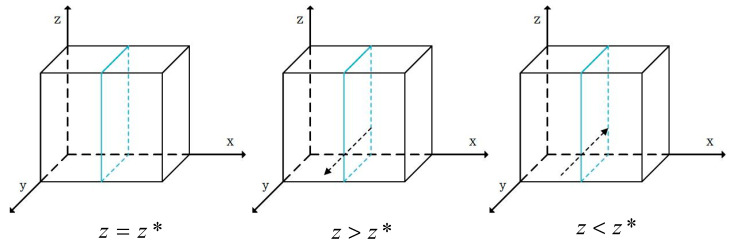
Phase diagrams of the evolution of slaughtering behaviors of pig slaughterers.

**Figure 6 foods-13-02387-f006:**
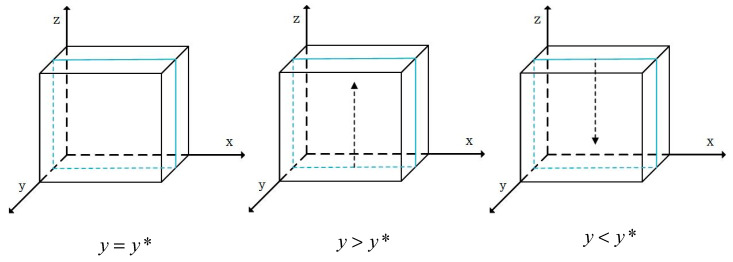
Phase diagrams of the evolution of the processing behaviors of pork manufacturers.

**Figure 7 foods-13-02387-f007:**
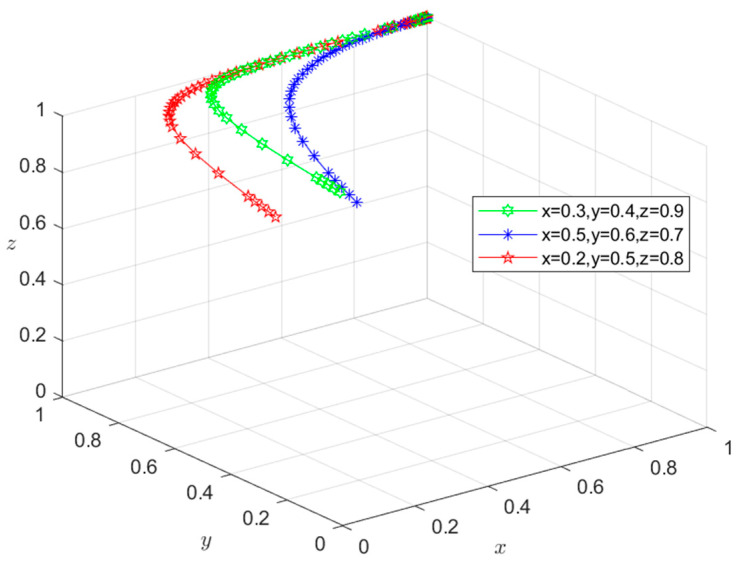
Stability test of equilibrium point *M*_1_(1,1,1).

**Figure 8 foods-13-02387-f008:**
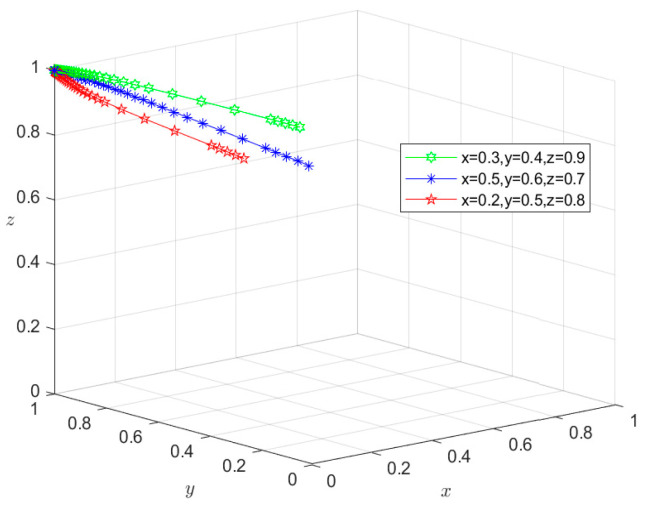
Stability test of equilibrium point *M*_2_(0,1,1).

**Figure 9 foods-13-02387-f009:**
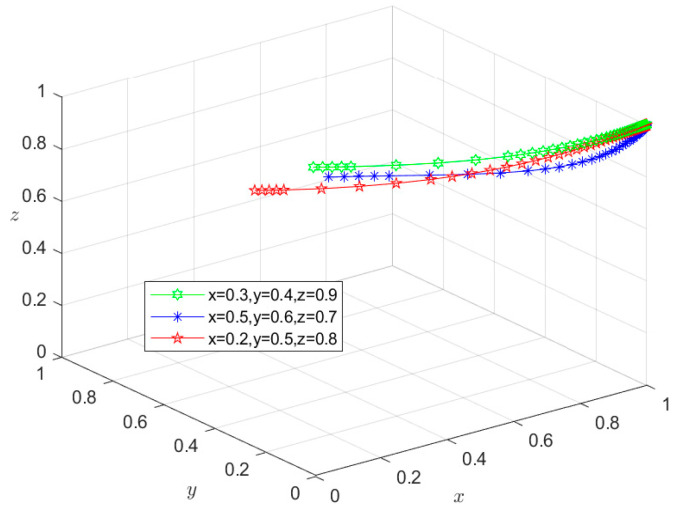
Stability test of equilibrium point *M*_3_(1,0,1).

**Figure 10 foods-13-02387-f010:**
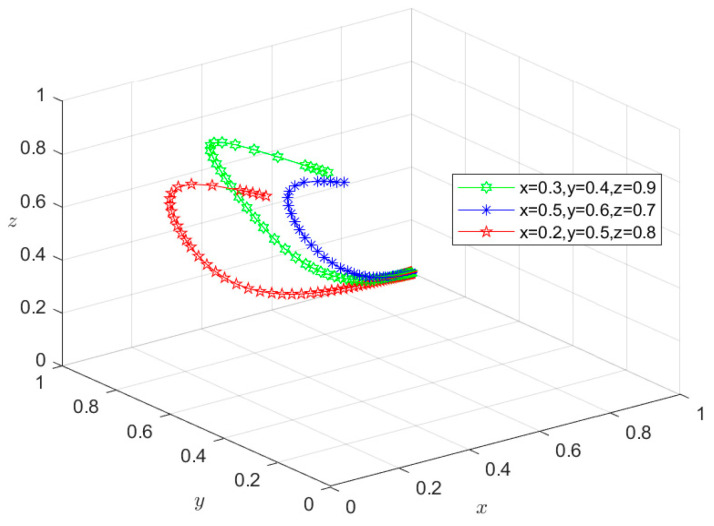
Stability test of equilibrium point *M*_4_(1,1,0).

**Figure 11 foods-13-02387-f011:**
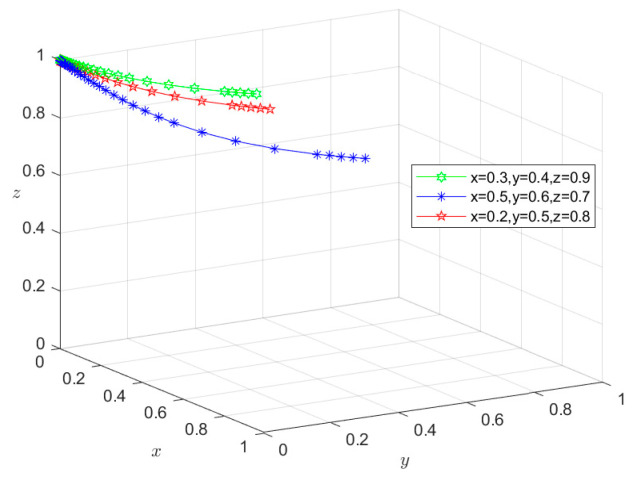
Stability test of equilibrium point *M*_5_(0,0,1).

**Figure 12 foods-13-02387-f012:**
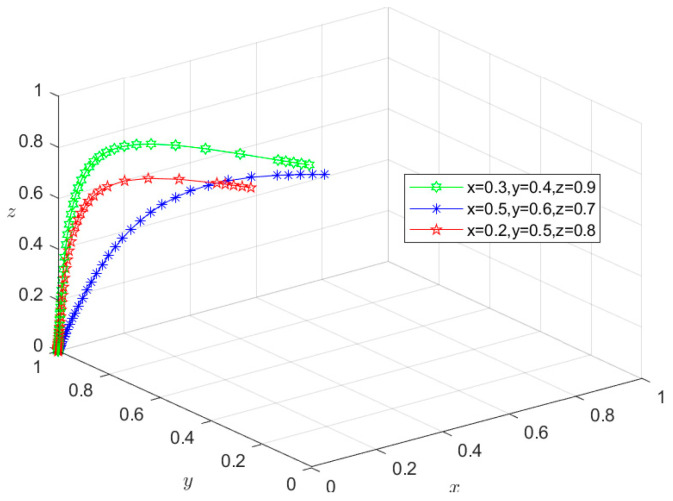
Stability test of equilibrium point *M*_6_(0,1,0).

**Figure 13 foods-13-02387-f013:**
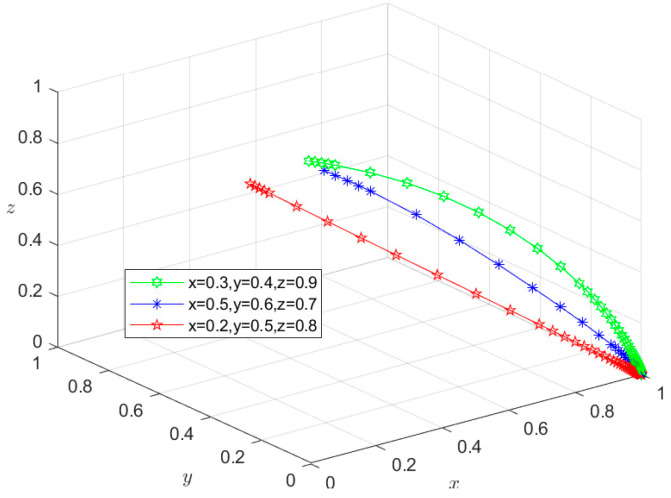
Stability test of equilibrium point *M*_7_(1,0,0).

**Figure 14 foods-13-02387-f014:**
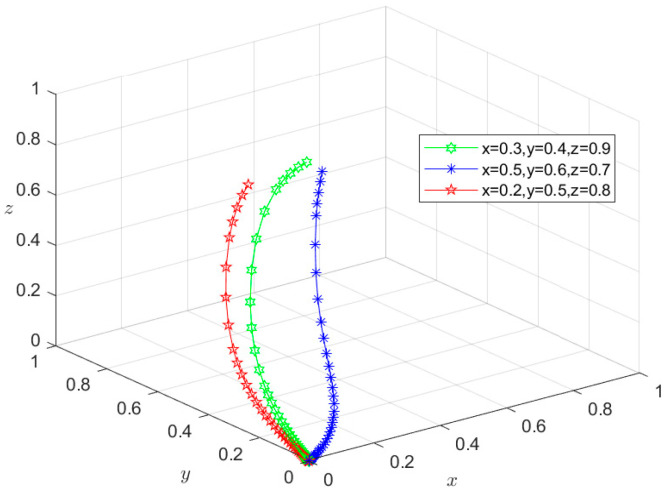
Stability test of equilibrium point *M*_8_(0,0,0).

**Figure 15 foods-13-02387-f015:**
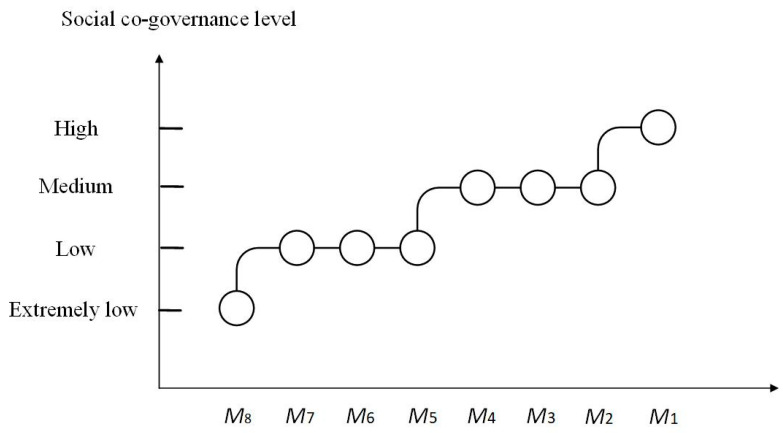
Schematic diagram of the mutual relationship between equilibrium points and social co-governance level.

**Figure 16 foods-13-02387-f016:**
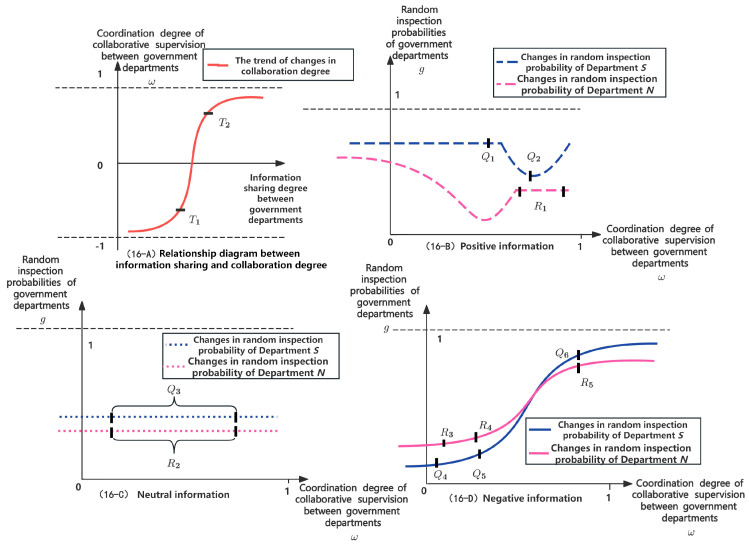
Schematic diagrams of the relationship between information sharing types and random inspection probabilities of Departments *N* and *S*.

**Figure 17 foods-13-02387-f017:**
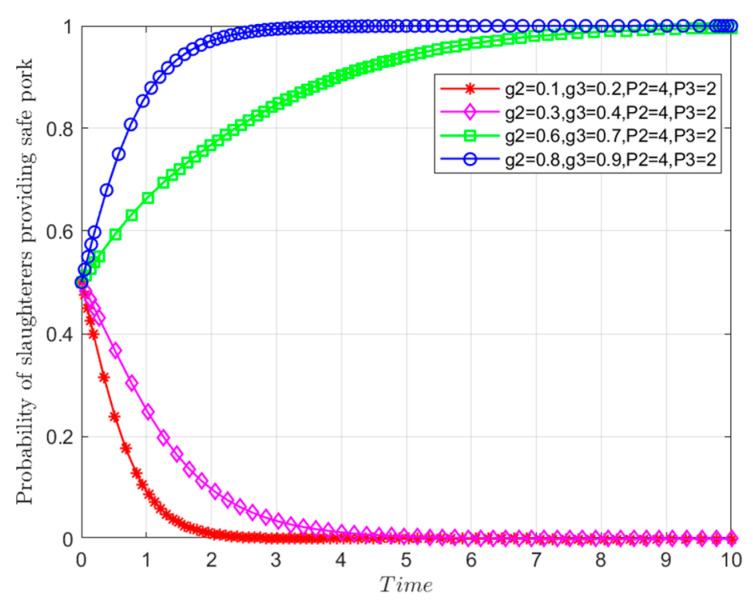
Influences of random inspection probabilities of government departments based on information sharing on the evolutionary paths of the behavioral strategy of *Y*.

**Figure 18 foods-13-02387-f018:**
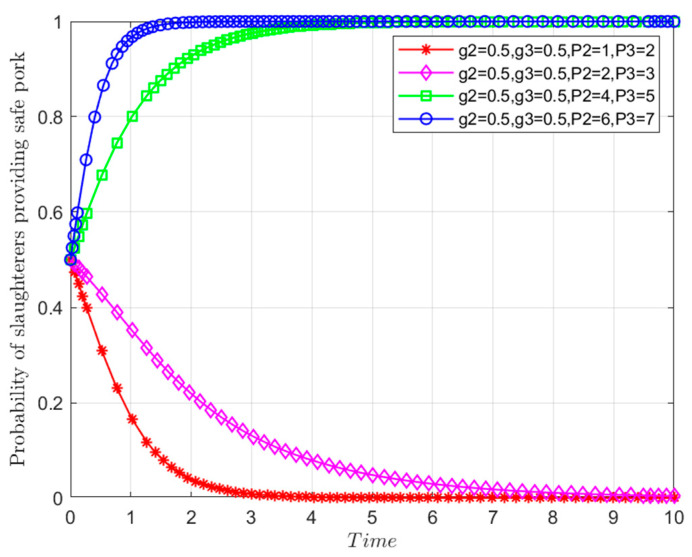
Influences of government economic punishment on the evolutionary path of the behavioral strategy of *Y*.

**Table 1 foods-13-02387-t001:** Parameters and their definitions.

Parameter Symbols	Definitions
*S* _1_	Expected revenue obtained by *X* from safe breeding
*S* _2_	Expected revenue obtained by *X* from risky breeding
*S* _3_	Expected revenue obtained by *Y* from safe slaughtering
*S* _4_	Expected revenue obtained by *Y* from risky slaughtering
*S* _5_	Expected revenue obtained by *Z* from safe processing
*S* _6_	Expected revenue obtained by *Z* from risky processing
ω	Coordination degree of collaborative supervision of Departments *N* and *S*
*g* _1_	Probability of Department *N*’s random inspection of *X*
*g* _2_	Probability of Department *N*’s random inspection of *Y*
*g* _3_	Probability of Department *S*’s random inspection of the pork slaughtered by *Y* in the market for sale
*g* _4_	Probability of Department *S*’s random inspection of *Z*
*P* _1_	Economic penalty on *X* from Department *N*
*P* _2_	Economic penalty on *Y* from Department *N*
*P* _3_	Economic penalty on *Y* from Department *S*
*P* _4_	Economic penalty on *Z* from Department *S*
*m* _1_	Probability of *Y* inspecting *X*
*m* _2_	Probability of consumers reporting *Y*
*m* _3_	Probability of *Z* inspecting *Y*
*m* _4_	Probability of consumers reporting *Z*
*m* _5_	Probability of consumers becoming aware of the market credit information of *Z* publicized by Department *G*
*H* _1_	Liquidated damage paid by *X* to *Y*
*H* _2_	Liquidated damage paid by *Y* to consumers
*H* _3_	Liquidated damage paid by *Y* to *Z*
*H* _4_	Compensation paid by *Z* Z to consumers
*H* _5_	Economic loss caused by the negative market reputation of *Z*
x	Probability of *X* choosing safe breeding, 0 ≤ x ≤ 1
y	Probability of *Y* choosing safe slaughtering, 0 ≤ y ≤ 1
z	Probability of *Z* choosing safe processing, 0 ≤ z ≤ 1

**Table 2 foods-13-02387-t002:** Payoff matrix of the trilateral game.

Pig Farmers	Pork Processing Manufacturers	Pig Slaughterers	
		Safe Slaughtering (y)	Risky Slaughtering (1 − y)
Safe breeding (x)	Safe processing (z)	*S* _1_	*S* _1_
		*S* _3_	*S*_4_ – *g*_2_*P*_2_ – *g*_3_*P*_3_ – *m*_2_*H*_2_ – *m*_3_*H*_3_
		*S* _5_	*S*_5_ + *m*_3_*H*_3_
	Risky processing (1 − z)	*S* _1_	*S* _1_
		*S* _3_	*S*_4_ – *g*_2_*P*_2_ – *g*_3_*P*_3_ – *m*_2_*H*_2_
		*S*_6_ − *g*_4_*P*_4_ − *m*_4_*H*_4_ − *m*_5_*H*_5_	*S*_6_ − *g*_4_*P*_4_ − *m*_4_*H*_4_ − *m*_5_*H*_5_
Risky breeding (1 − x)	Safe processing (z)	*S*_2_ − *g*_1_*P*_1_ – *m*_1_*H*_1_	*S*_2_ − *g*_1_*P*_1_
		*S*_3_ + *m*_1_*H*_1_	*S*_4_ – *g*_2_*P*_2_ – *g*_3_*P*_3_ – *m*_2_*H*_2_ – *m*_3_*H*_3_
		*S* _5_	*S*_5_ + *m*_3_*H*_3_
	Risky processing (1 − z)	*S*_2_ − *g*_1_*P*_1_ – *m*_1_*H*_1_	*S*_2_ − *g*_1_*P*_1_
		*S*_3_ + *m*_1_*H*_1_	*S*_4_ – *g*_2_*P*_2_ – *g*_3_*P*_3_ – *m*_2_*H*_2_
		*S*_6_ − *g*_4_*P*_4_ − *m*_4_*H*_4_ − *m*_5_*H*_5_	*S*_6_ − *g*_4_*P*_4_ − *m*_4_*H*_4_ − *m*_5_*H*_5_

**Table 3 foods-13-02387-t003:** Characteristic values of system equilibrium points and stability conditions.

Equilibrium Points	Characteristic Values			Stability Conditions
	λ*_i_*_1_	λ*_i_*_2_	λ*_i_*_3_	
*M*_1_(1,1,1)	$S_{2}-S_{1}-g_{1}P_{1}-m_{1}H_{1}$	$\begin{array}{c} S_{4}-S_{3}-g_{2}P_{2}-g_{3}P_{3}\\ -m_{2}H_{2}-m_{3}H_{3} \end{array}$	$\begin{array}{c} S_{6}-S_{5}-g_{4}P_{4}\\ -m_{4}H_{4}-m_{5}H_{5} \end{array}$	Condition (1)
*M*_2_(0,1,1)	$S_{1}-S_{2}+g_{1}P_{1}+m_{1}H_{1}$	$\begin{array}{c} S_{4}-S_{3}-g_{2}P_{2}-g_{3}P_{3}\\ -m_{1}H_{1}-m_{2}H_{2}-m_{3}H_{3} \end{array}$	$\begin{array}{c} S_{6}-S_{5}-g_{4}P_{4}\\ -m_{4}H_{4}-m_{5}H_{5} \end{array}$	Condition (2)
*M*_3_(1,0,1)	$S_{2}-S_{1}-g_{1}P_{1}$	$\begin{array}{c} S_{3}-S_{4}+g_{2}P_{2}+g_{3}P_{3}\\ +m_{2}H_{2}+m_{3}H_{3} \end{array}$	$\begin{array}{c} S_{6}-S_{5}-g_{4}P_{4}\\ -m_{3}H_{3}-m_{4}H_{4}-m_{5}H_{5} \end{array}$	Condition (3)
*M*_4_(1,1,0)	$S_{2}-S_{1}-g_{1}P_{1}-m_{1}H_{1}$	$\begin{array}{l} S_{4}-S_{3}-g_{2}P_{2}\\ -g_{3}P_{3}-m_{2}H_{2} \end{array}$	$\begin{array}{c} S_{5}-S_{6}+g_{4}P_{4}\\ +m_{4}H_{4}+m_{5}H_{5} \end{array}$	Condition (4)
*M*_5_(0,0,1)	$S_{1}-S_{2}+g_{1}P_{1}$	$\begin{array}{c} S_{3}-S_{4}+g_{2}P_{2}+g_{3}P_{3}\\ +m_{1}H_{1}+m_{2}H_{2}+m_{3}H_{3} \end{array}$	$\begin{array}{c} S_{6}-S_{5}-g_{4}P_{4}\\ -m_{3}H_{3}-m_{4}H_{4}-m_{5}H_{5} \end{array}$	Condition (5)
*M*_6_(0,1,0)	$S_{1}-S_{2}+g_{1}P_{1}+m_{1}H_{1}$	$\begin{array}{c} S_{4}-S_{3}-g_{2}P_{2}\\ -g_{3}P_{3}-m_{1}H_{1}-m_{2}H_{2} \end{array}$	$\begin{array}{c} S_{5}-S_{6}+g_{4}P_{4}\\ +m_{4}H_{4}+m_{5}H_{5} \end{array}$	Condition (6)
*M*_7_(1,0,0)	$S_{2}-S_{1}-g_{1}P_{1}$	$\begin{array}{l} S_{3}-S_{4}+g_{2}P_{2}\\ +g_{3}P_{3}+m_{2}H_{2} \end{array}$	$\begin{array}{c} S_{5}-S_{6}+g_{4}P_{4}\\ +m_{3}H_{3}+m_{4}H_{4}+m_{5}H_{5} \end{array}$	Condition (7)
*M*_8_(0,0,0)	$S_{1}-S_{2}+g_{1}P_{1}$	$\begin{array}{c} S_{3}-S_{4}+g_{2}P_{2}+g_{3}P_{3}\\ +m_{1}H_{1}+m_{2}H_{2} \end{array}$	$\begin{array}{c} S_{5}-S_{6}+g_{4}P_{4}\\ +m_{3}H_{3}+m_{4}H_{4}+m_{5}H_{5} \end{array}$	Condition (8)
Condition (i): λ*_i_*_1,_ λ*_i_*_2,_ λ*_i_*_3_ < 0				

**Table 4 foods-13-02387-t004:** Equilibrium points and assignment of each parameter.

Parameters	Equilibrium Points							
	*M*_1_(1,1,1)	*M*_2_(0,1,1)	*M*_3_(1,0,1)	*M*_4_(1,1,0)	*M*_5_(0,0,1)	*M*_6_(0,1,0)	*M*_7_(1,0,0)	*M*_8_(0,0,0)
*S* _1_	6	6	6	6	6	6	6	6
*S* _2_	8	10	8	8	8	10	8	8
*S* _3_	4	4	4	4	4	4	4	4
*S* _4_	6	8	12	8	10	6	10	10
*S* _5_	4	4	4	4	4	4	4	4
*S* _6_	6	6	8	10	8	10	10	10
*g* _1_	0.5	0.5	0.5	0.5	0.5	0.5	0.5	0.5
*g* _2_	0.5	0.5	0.5	0.5	0.5	0.5	0.5	0.5
*g* _3_	0.5	0.5	0.5	0.5	0.5	0.5	0.5	0.5
*g* _4_	0.5	0.5	0.5	0.5	0.5	0.5	0.5	0.5
*P* _1_	2	2	6	2	2	2	6	2
*P* _2_	2	4	2	4	2	4	4	4
*P* _3_	4	4	4	4	2	4	4	4
*P* _4_	4	4	4	4	4	4	4	4
*m* _1_	0.5	0.5	0.5	0.5	0.5	0.5	0.5	0.5
*m* _2_	0.5	0.5	0.5	0.5	0.5	0.5	0.5	0.5
*m* _3_	0.5	0.5	0.5	0.5	0.5	0.5	0.5	0.5
*m* _4_	0.5	0.5	0.5	0.5	0.5	0.5	0.5	0.5
*m* _5_	0.5	0.5	0.5	0.5	0.5	0.5	0.5	0.5
*H* _1_	4	2	2	4	2	2	2	2
*H* _2_	4	4	4	4	2	4	4	4
*H* _3_	2	2	2	2	2	2	2	2
*H* _4_	2	2	2	2	2	2	2	2
*H* _5_	2	2	2	2	2	2	2	2

## Data Availability

The original contributions presented in the study are included in the article, further inquiries can be directed to the corresponding author.

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
