# Peer review of "Study on the Influences of Inter-Governmental Information Flow and Interdepartmental Collaborative Supervision on Pork Quality: A Case Study in China"

_foods, 2024, doi:10.3390/foods13152387_

Round 1
Reviewer 1 Report
Comments and Suggestions for Authors
The paper is very interesting and well developed. I have just some recommendations:
1. In the abstract, I believe you could make the objective clearer, for example: “Based on a social co-governance framework, this study established a three-party evolutionary game model involving pig farmers, pig slaughterers, and pork processing manufacturers”… for what?
2. Introduction: It is adequate to include at the end of the introduction a description of the sections that come ahead in the manuscript. It seems you have included this in the literature review section (it should be in the introduction).
3. In sections 4 and 5, it is interesting to cite references that used the same variables and models.
4. In the conclusion section, it is recommended to compare the results of this study to other similar studies and include some discussion about the contribution of the paper to this research theme and practice issues in the application area.
Author Response
- In the abstract, I believe you could make the objective clearer, for example: “Based on a social co-governance framework, this study established a three-party evolutionary game model involving pig farmers, pig slaughterers, and pork processing manufacturers”… for what?
Author response: Thank you for your valuable suggestions. The abstract has been modified according to your comments. Please see Line 8-10 of the revised manuscript.
- Introduction: It is adequate to include at the end of the introduction a description of the sections that come ahead in the manuscript. It seems you have included this in the literature review section (it should be in the introduction).
Author response:Thank you for your suggestions. We have re-arranged the content in question according to your suggestions.Please see Line 75-102 of the revised manuscript.
- In sections 4 and 5, it is interesting to cite references that used the same variables and models.
Author response:Thank you.The assumed parameters (variables) of the model in this paper were all set based on literature research and the realities specific to China.
- In the conclusion section, it is recommended to compare the results of this study to other similar studies and include some discussion about the contribution of the paper to this research theme and practice issues in the application area.
Author response:Thank you for your valuable suggestions. We have incorporated comparisons with existing literature conclusions based on your feedback. Please see Line 740, Line 744, Line 747-748, Line 753 and Line 758-759 of the revised manuscript.
The contributions of our study were outlined in the original version of our manuscript but this explanation was not comprehensive. We have made further improvements based on your suggestions. Please see Line 100-102 and Line 779-789 of the revised manuscript.
The practical issues related to the application field of this paper are addressed to some extent in Section 8 ("Conclusions and Policy Implications"). Based on your suggestions, we have further refined the wording to enhance clarity. Please see Line 760 of the revised manuscript.
Reviewer 2 Report
Comments and Suggestions for Authors
Overall, the paper meets high standards of scientific writing and research. However, I would like to draw the authors' attention to two areas that could benefit from slight improvement:
1. Provide detailed criteria for selecting pig farmers, slaughterers, and pork manufacturers for the study. This will help readers better understand how the study participants were chosen and enhance the overall transparency of the methodology.
2. Some text sections are dense and could benefit from simplification. Aim for clarity and conciseness in explanations and descriptions to make the manuscript more accessible to a broader audience. Avoid unnecessary jargon and overly complex sentences.
Good luck!
Comments on the Quality of English LanguageThe English is generally clear, but there are some areas where minor grammatical and stylistic improvements could be made to enhance readability.
Author Response
- Provide detailed criteria for selecting pig farmers, slaughterers, and pork manufacturers for the study. This will help readers better understand how the study participants were chosen and enhance the overall transparency of the methodology.
Author response:Thank you for your suggestions, We have modified the manuscript according to your suggestions and added a new figure in the updated version of our manuscript. Please see Line 228-238 of the revised manuscript.
- Some text sections are dense and could benefit from simplification. Aim for clarity and conciseness in explanations and descriptions to make the manuscript more accessible to a broader audience. Avoid unnecessary jargon and overly complex sentences.
Author response:Thank you for your suggestions. We have made numerous changes to address the linguistic organization issues and have worked to make the content more accessible and understandable. However, we would like to note that the research in this paper relies on a three-party game model, and the mathematical formulas and derivation processes are inherently technical and involve many specialized terms, which may be challenging for most audiences to fully grasp. Please see Line 248-251, Line 270-279, Line 303-306, Line 323-324, Line 345-347 and Line 523-525 of the revised manuscript.
Reviewer 3 Report
Comments and Suggestions for Authors
This is a good paper. It has well described methods and results. In some aspects it should be improved.
1. Abstract should include methods.
2. Introduction should be improved. Please write aim, research questions and how the paper is organized. What od the added value added is not clear.
3. Small correction is need. Page 8 line 19 from top.
4. Before conclusion should be discusion section. In such section authors should discus their research with other authors research. That is why when you write such section you will have to extent your literature.
5. I can see no sources under tables and figures. In my opiniom you should write for example own elaboration based on...
Comments on the Quality of English LanguageEnglish is good but on some areas it needs small correction.
Author Response
- Abstract should include methods.
Author response:Thank you for your suggestion. We have added research methods to the revised abstract.Please see Line 10-13 of the revised manuscript.
- Introduction should be improved. Please write aim, research questions and how the paper is organized. What od the added value added is not clear.
Author response:Your suggestions are very valuable. We have made modifications one by one according to your suggestions. Please see Line 68-70, Line 70-72, and Line 75-102 of the revised manuscript.
Your comments on the contributions of this paper are similar to those of Reviewer #1. We have revised the statements regarding the paper’s contribution based on your suggestions. Please see Line 100-102 and Line 779-789 of the revised manuscript.
- Small correction is need. Page 8 line 19 from top.
Author response:We have made the necessary modifications based on your suggestion. Please see Line 323-324 of the revised manuscript.
- Before conclusion should be discussion section. In such section authors should discus their research with other authors’ research. That is why when you write such section you will have to extent your literature.
Author response:Thank you for your suggestions and your suggestions are similar to those of Reviewer #1.Please see Line 740, Line 744, Line 747-748, Line 753 and Line 758-759 of the revised manuscript.
- I can see no sources under tables and figures. In my opinionyou should write for example own elaboration based on...
Author response:Thank you. The data displayed in the tables and figures of this paper are derived from parameter assumptions and model calculations within the game model, rather than from surveys or reference data from other literature. Therefore, it is not necessary to declare a source for this data.
Round 2
Reviewer 3 Report
Comments and Suggestions for Authors
The authors responded to my comments and improved the paper. IT can be published after checking English.
Comments on the Quality of English LanguageAs above. Paper Has been improved and can be published. Small English check is recommended.